# A general method for the creation of dilational surfaces

Freek G.J. Broeren [1,2]*, Werner W.P.J. van de Sande[1,2], Volkert van der Wijk[1] & Just L. Herder [1]

Dilational structures can change in size without changing their shape. Current dilational designs are only suitable for specific shapes or curvatures and often require parts of the structure to move perpendicular to the dilational surface, thereby occupying part of the enclosed volume. Here, we present a general method for creating dilational structures from arbitrary surfaces (2-manifolds with or without boundary), where all motions are tangent to the described surface. The method consists of triangulating the target curved surface and replacing each of the triangular faces by pantograph mechanisms according to a tiling algorithm that avoids collisions between neighboring pantographs. Following this algorithm, any surface can be made to mechanically dilate and could, theoretically, scale from the fully expanded configuration down to a single point. We illustrate the method with three examples of increasing complexity and varying Gaussian curvature.

---

[1] Department of Precision and Microsystems Engineering, Faculty of Mechanical, Maritime and Materials Engineering, Delft University of Technology, Mekelweg 2, 2628 CDDelft, The Netherlands. [2] These authors contributed equally: Freek G. J. Broeren, Werner W. P. J. van de Sande. *email: f.g.j. broeren@tudelft.nl

Expandable structures are of significant relevance in nature and engineering and come in a variety of forms. Natural examples include the stowing of the precious wings of beetles[1] or the fitting of young leaves into buds[2]. Numerous engineering examples can be found as well, including satellite antennas and solar panels that need to be compact for launch and expand for operation[3–6], and medical stents that need to be moved through arteries[7] or the esophagus[8] in compacted form and deploy at the target position.

Most expandable structures rely on an underlying mechanism to allow them to be reversibly compacted. One well-known example of an expandable structure in which the mechanism is clearly visible is the Hoberman Sphere[9,10]. This mechanism dilates, i.e., its envelope changes in size without changing its shape[11,12]. We define dilational structures as structures composed of mechanisms whose only degree of freedom corresponds to dilation. Other examples of dilational structures are dilational polyhedral linkages[13,14].

Dilational structures have also been studied in the field of mechanical metamaterials[15,16], particularly for auxetic behavior where the Poisson's ratio is negative[17–19]. A Poisson's ratio of exactly -1 corresponds to dilation.

Currently, several limitations in dilational structures exist. Firstly, most dilational structures have been designed for a specific shape or curvature, making these mechanisms applicable to a very limited set of shapes; for example, the buckliball[20], which is based on a polyhedral linkage that resembles a sphere. Secondly, linkages such as the Hoberman Sphere use mechanisms that move perpendicular to the described surface, making them protrude into the enclosed volume, which, for instance, could be a problem in stent design. Thirdly, to the authors knowledge, no examples of spatial mechanical metamaterials exist that can be sculpted to thin, curved surfaces. The unit cell that governs the behavior of such structures inherently has a finite volume, making the construction of thin dilational surfaces with only motion tangent to the plane impossible. Also, the current planar auxetic metamaterials can not, in general, be used, since the underlying kinematics are not valid for arbitrarily curved surfaces.

In this paper, we present a general method to create mechanism-based dilational structures fitting to any spatially curved surface, by which we mean 2-manifolds with or without boundary. Our method improves on existing work on two key points. Firstly, the resulting mechanism structure is placed on the surface, with no parts of the mechanism moving into the enclosed volume and normal to the surface, unlike for instance the Hoberman mechanism. Secondly, the method is applicable to surfaces with any curvature and can even be applied to non-closed surfaces, i.e., surfaces containing holes or cuts. Enabling these properties in dilational structures makes them of use in, for instance, structures that grow with a person such as medical braces for children and expandable furniture, medical devices that require stowability or compression but need to be stiff otherwise, or implants that need to accommodate some motion but maintain their shape, such as aortic stents.

In the following, we describe the method, where we first triangulate the surface and then place pantograph mechanisms on each of the faces of the triangulation. We prove that this method can be used for any spatially curved surface and comment on the maximum scaling factor possible for these structures. Finally, we apply our method to three surfaces of increasing complexity, illustrating its versatility.

## Results

**Dilation**. Dilation is a homothetic transformation that relates two similar shapes with respect to a homothetic center[21]. Any two figures related by a dilation are similar and have the same orientation (see Fig. 1). This transformation preserves the shape and orientation of the figure, but changes the size of each of the elements of the structure by the same factor. In a dilational structure the distances between a representative set of points on the structure, typically corner joints, all scale by the same factor during actuation.

**Triangulation**. The first step of the presented design method is to triangulate the curved surface from which we want to create a dilational structure. Triangulation is a common strategy to approximate curved surfaces by a mesh of triangular faces and lies at the basis of the STL file format used in 3D printing[22,23]. Triangulation is illustrated in Fig. 2 for a sphere. It is observed that the accuracy of the approximation increases with the number of triangular faces in the resulting mesh.

The triangulation results in a polyhedral surface with only triangular faces. It can be shown that every surface (by which we mean a 2-manifold with or without boundary) can be triangulated such that at most two triangular faces share an edge[24]. If the resulting polyhedron undergoes dilation, the number of triangles, their shapes and their respective relations must stay constant. Only the size of the triangles is allowed to change, their aspect ratio and orientation are preserved.

**Pantograph linkage**. To transform the polyhedral surface into a movable linkage with dilational motion, we employ the skew, or Sylvester's, pantograph mechanism[25–27].

The skew pantograph is a four-bar mechanism of which two adjacent links are extended into triangles. The mechanism has revolute joints at $p$, $q$, $C$ and $r$, as illustrated in Fig. 3. The link $Cq$ is equal in length to side $rp$, as are $Cr$ and $qp$, which makes $pqCr$ a parallelogram. Also, the triangles $Apq$ and $pBr$ are similar. A resulting feature from these properties is that in any pose of the mechanism the triangle $ABC$ is similar to triangles $Apq$ and $pBr$. The proof can be found in Hall (1961)[28]. When the mechanism moves, the distances $AC$, $AB$, and $BC$ become smaller as the parallelogram decreases in area. As a result, the mechanism has a single degree of freedom and during motion the striped red triangle described by its three similarity points (indicated $A$, $B$, and $C$) changes in size but remains similar of shape, as is illustrated for three poses. These three joints will be referred to as the similarity points of the pantograph linkage.

The maximum scaling that can be achieved with a skew pantograph depends on the placement of the joints on the edges of the spanned triangle in the neutral position. We place the joint in the middle of the sides of the spanned triangle in the neutral

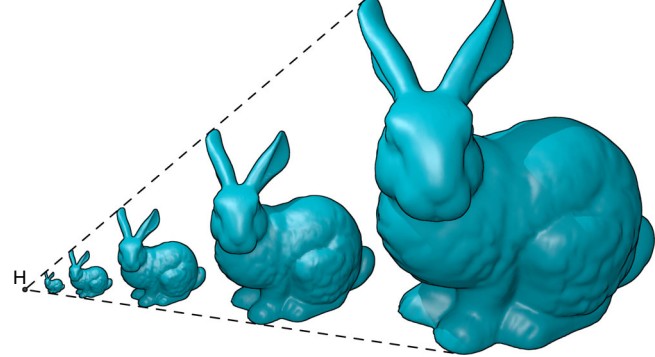

**Fig. 1** Dilation of structures. Under dilation, a structure scales with respect to a homothetic center (point H in this figure), preserving its size and orientation

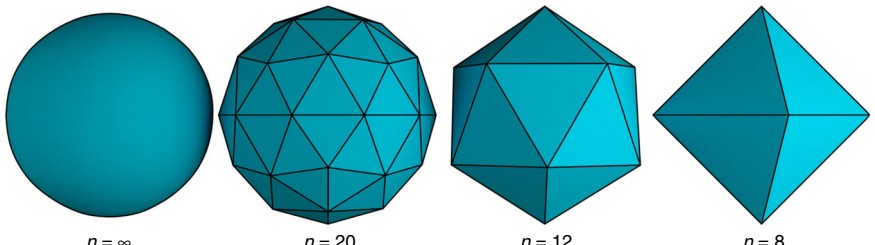

**Fig. 2** Triangulation of surfaces. Any curved surface, in this case a sphere, can be approximated by a mesh of triangles in a process called triangulation. A larger mesh of triangular faces gives a better approximation of the original surface. Shown are polyhedral triangulations of a sphere with $n$ triangular faces

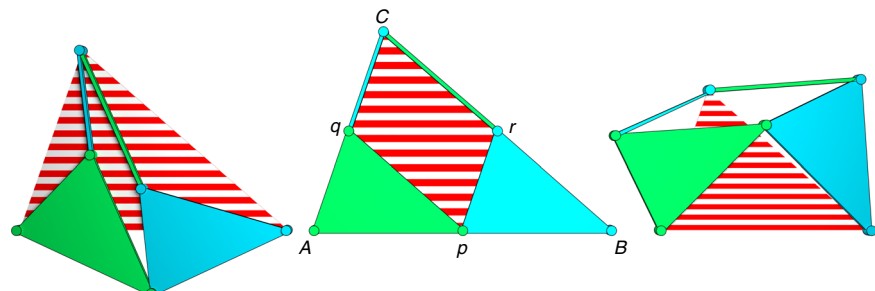

**Fig. 3** The skew pantograph. The skew pantograph is a one-degree-of-freedom mechanism that scales the spanned triangle (i.e., the red-striped area) determined by three of its joints; the similarity points, indicated $A$, $B$, and $C$. This mechanism has revolute joints at points $p$, $q$, $C$, and r. For the neutral position, shown in the middle, the spanned triangle equals the triangular shape of the pantograph, this is a useful pose for constructing dilational surfaces

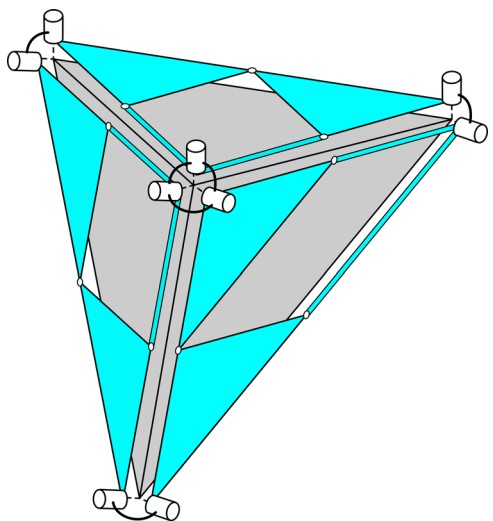

**Fig. 4** Connections between pantographs. The relative orientation of adjacent pantograph mechanisms is maintained by compound universal joints. This figure shows part of a dilational surface, the triangular faces are shown in gray, and the rigid parts of the pantograph mechanisms are shown in blue. At each of the corners of the pantographs, they are coupled to their neighbors by universal joints. A universal joint consists of two revolute joints (shown as white cylinders) in series. A compound universal joint is created at an intersection with three (or more) triangular faces

The pantograph mechanisms ensures the proper scaling of each individual face. However, in order for the whole structure to dilate, it is also required that each face simultaneously scales by the same factor and that the faces do not rotate with respect to each other. We achieve this by connecting neighboring pantograph mechanisms by means of compound universal joints, whose description follows.

Two adjacent faces of the triangulated surface share a single edge and two vertices. In order to maintain the mobility of the neighboring pantograph mechanisms, they can only be connected at the vertices. At these points, we connect the pantograph mechanisms with universal joints (two consecutive revolute pairs), of which the axes are parallel to the normals of the respective faces, as is illustrated in Fig. 4. This configuration constrains the rotation about the shared edge of the faces and therefore preserves the relative orientation of the two faces.

Because the two neighboring pantograph mechanisms are now connected along the shared edge, their degrees of freedom are also coupled. When one of the pantograph mechanisms moves, the length of the shared edge will change, causing a movement in the other pantograph. In this way, it is ensured that both pantographs are scaled simultaneously by the same factor and maintain their relative orientation.

Each set of neighboring pantograph mechanisms is connected in this way, creating compound universal joints at the corners of the faces. This preserves the relative angles of all faces and ensures the same scale factor for each of the faces. Therefore, the total resulting motion will be dilation.

position, such that $Aq = qC$. In this way, the rigid triangles are sized down by a factor 2 relative to the spanned triangle, which allows the spanned triangle to scale down to a single point.

**Coupling the pantographs.** Each of the faces obtained by triangulating the curved surface is replaced by a skew pantograph mechanism. In this way, we ensure that each individual face can only deform by scaling, keeping its original shape.

**Range of motion of the pantograph.** Kinematically, a structure constructed from the proposed pantograph mechanisms can scale down to a point. In reality, the range of motion of the planar pantographs is limited because of collisions among the rigid bodies that make up the pantographs. In this section, we highlight the factors limiting the range of motion of the dilational mechanisms constructed from pantographs and discuss how to minimize their effects.

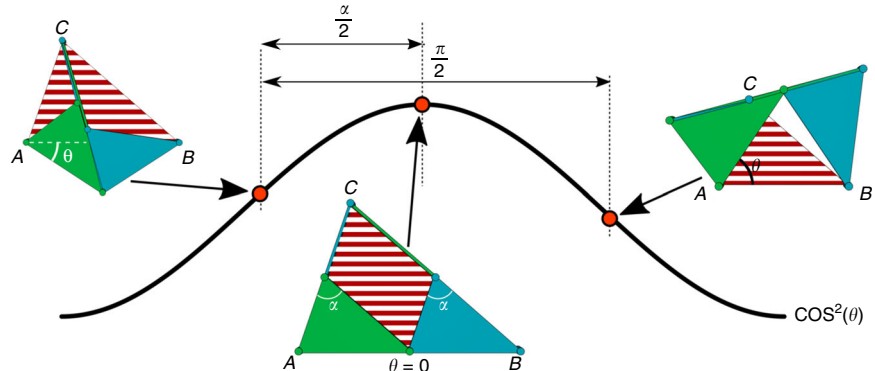

**Fig. 5** Range of motion of the skew pantograph. The limits of scaling of a skew pantograph. The area of the spanned triangle (shown as a red-striped area) scales with $\cos^2(\theta)$. The red dots illustrate the values of $\theta$ in each of the drawings. The end of motion is reached when the bars of the linkage become collinear (left and right drawings). Between these states, the rotation angle of $\theta$ is always $\frac{\pi}{2}$. The distribution of this range over the left and right motion directions depends on the top angle $\alpha$

We have described the pantograph mechanism used to make the triangular faces of the polygons dilational. The motion of the mechanism can be described by a single parameter $\theta$ and the sides of the spanned triangle are scaled by a factor $\lambda = \cos(\theta)$ when the mechanism is actuated[29]. The area of the spanned triangle is then scaled by a factor of $\lambda^2 = \cos^2(\theta)$, see Fig. 5.

Starting at the neutral position, where $\theta = 0$, the mechanism can move in two directions: $\theta$ can increase or decrease, corresponding to a counter-clockwise or clockwise rotation of the lower left (green) rigid triangle. In both cases, the effect on the scaling of the spanned triangle will be the same, since $\cos(\theta)$ is symmetric around $\theta = 0$.

In the case where $\theta$ increases, the mechanism will protrude out of the spanned triangle at two edges, while it will open up free space at the third edge. Conversely, when $\theta$ decreases, the mechanism protrudes out of the spanned triangle at one edge, and opens up space at the other two edges.

When the rigid bodies of the pantograph mechanism are allowed to overlap and cross each other, the minimum area of the spanned triangle is obtained at $\theta = \pm\frac{\pi}{2}$ for the two different cases, both resulting in a scaling factor $\lambda = 0$. When collisions are considered, these values of $\theta$ can no longer be reached and often, the scaling factor $\lambda$ differs between the two motion directions.

If the pantographs are designed to be planar and therefore move within a single plane, the range of motion is limited by internal collisions of the links. All pantograph mechanisms in the dilational surface are linked to have a single degree of freedom. Therefore, when one pantograph is actuated such that, for that mechanism, we obtain a rotation $\theta$ in its triangular faces, all other pantograph mechanisms in the dilational surface will have a rotation of $\pm\theta$. This causes the complete mechanism to reach the end of its motion as soon as self-collision occurs in any pantograph of the structure. Therefore, this is the main limiting factor on the maximum scaling factor of the assembled dilational mechanism.

Looking at Fig. 5, we can see that the pantograph mechanism reaches the end of its range of motion when $\theta = -\frac{\alpha}{2}$ for one direction of motion, or when $\theta = \frac{\pi-\alpha}{2}$ for the other direction, where $\alpha$ is the top angle of the spanned triangle. At these points, the binary links of the pantograph mechanism become collinear. In this calculation, we have considered the links as lines of zero width. In reality, the links and joints from which the pantograph mechanisms are constructed will have finite width. This will cause collisions to happen earlier and the range of motion to be limited further.

The total range of $\theta$ is $\frac{\pi}{2}$ radians, because the internal angles between the links at two adjacent corners of a parallelogram four-bar linkage always add up to $\pi$. For the case where the links have

zero width, the optimal scaling factor would be found when $\alpha = 0$ or when $\alpha = \pi$, allowing only for motion in one direction.

However, in both of these cases, the pantograph degenerates to a line, in which case no feasible mechanism would be possible. For simplicity and ease of tiling, it is beneficial when both motions directions have the same range from the neutral position. This is the case for $\alpha = \frac{\pi}{2}$; i.e., when the pantograph mechanisms are right-angled. In this case, the maximal scaling factor is $\lambda = \cos\left(\frac{\pi}{4}\right) = \frac{\sqrt{2}}{2} \approx 0.71$. So, when self-collisions are considered, the distances between points on the dilational surface can be scaled down by at most 29% relative to the neutral position.

**Placement of the pantographs**. When the pantograph mechanism moves, some parts of the mechanism protrude out of the spanned triangle, while other parts move into the spanned triangle. When all the faces of a triangulated surface are replaced by pantograph mechanisms, two neighboring pantographs could have parts moving out of the respective spanned triangles at their shared edge. This will cause neighboring pantographs to collide, locking the motion of the structure and thereby no longer allowing the scaling of the structure. To remedy this, we have created an algorithm that places the pantographs on a triangulated surface such that neighboring pantographs move along with each other, i.e., when one side of a pantograph has parts that move out of the spanned triangle, the corresponding side of the neighboring pantograph will have parts moving inwards. The algorithm consists of the following procedure.

We first construct the dual graph to the triangulated surface. In this graph, there is one node for each triangular face and two nodes are connected if the two corresponding faces share an edge. Such a graph is shown in Fig. 6a for the octahedron. Note that, as was mentioned in Section "Triangulation", at most two faces share an edge since the original surface is a 2-manifold. The dual graph to an octahedron is shown in Fig. 6a. For closed surfaces, this creates a simple, connected, 3-regular graph. We assign a direction to each of the edges in the graph to represent the motions of the pantograph mechanisms placed on the triangulated surface. A directed dual graph for an octahedron is shown in Fig. 6b. Since each edge in the dual graph can only have a single direction, the sides of the triangles are enforced to move along with each other.

The movement of the pantograph mechanisms is such that either the links on two sides move out of the spanned triangle and the links on the other into it or vice-versa, as shown in Fig. 3. Therefore, we require for each vertex of the graph that its

indegree and outdegree are larger than zero. In Supplementary Note 1, we show that for each simple connected graph where every node has a degree of at least 2, it is always possible to find an orientation of the graph such that both the indegree and outdegree of every node are larger than zero. The dual graph to every feasible triangulated surface always has nodes of degree at least 3; nodes with degree larger than three correspond to holes in the surface. Therefore, there must exist an orientation of the pantographs on each surface such that movement without collisions between the pantograph mechanisms is possible.

To find a suitable orientation, we use an algorithm that searches for flows through the representing graphs. A flow through a vertex ensures that the difference between the indegree and outdegree is one. This algorithm is further discussed in Supplementary Note 2. Once this orientation is found, the pantographs can be placed accordingly, see Figs. 6c, d for an example.

**Examples**. In this section, we will show the application of our method to three surfaces: an octahedron as a simple example[29], a cardioid with both positive and negative Gaussian curvature, and the Stanford Bunny as an advanced example. For all three examples, the reported maximum scaling is based on the pantographs in the resulting structure with the largest and smallest top angle, as described in subsection "Range of motion of the pantograph".

We start with the octahedron. This polygonal surface can be viewed as a very rough triangulation of a sphere, comprised of only 8 triangular faces. The eight faces of the octahedron are replaced by pantograph mechanisms, see Fig. 7 and

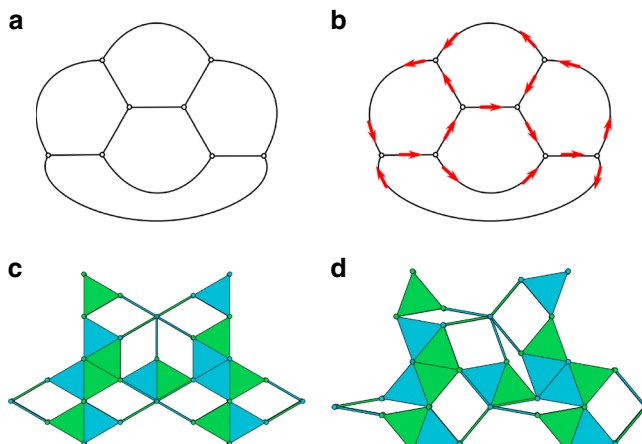

**Fig. 6** Tiling method. Steps in obtaining a correct tiling pattern of pantographs: **a** dual graph of the net, where the vertices are the faces of the triangulated surface, **b** directed dual graph with correct orientation, **c** correct tiling pattern, **d** correct tiling pattern with $\theta = 20°$

Supplementary Movie 1. In this way, a dilational surface with only equilateral faces is obtained ($\alpha = 60°$); this gives them a range of $[-30°, 60°]$. Any placement of the pantograph linkages will include pantographs with opposite motion directions, the maximum scaling can therefore be calculated to be $\lambda = \cos(30°) = 0.866$.

As a second example, we look at the cardioid. The cardioid is a planar curve obtained by tracing a point on a circle, which is rolled around a second circle with equal radius. This curve can be parameterized as follows:

$$\begin{aligned} x(t) &= a(2\cos(t) - \cos(2t)) \\ y(t) &= a(2\sin(t) - \sin(2t)). \end{aligned} \tag{1}$$

By revolving this curve around the $x$-axis, we obtain a spatial surface, as is shown in Fig. 8.

We have triangulated this shape by taking a planar map of the surface and performing a Delaunay triangulation[30,31] on this map. The points of the triangulation have been chosen to minimize the number of sharp angles in the triangulation. This triangulation is shown in Fig. 9a.

On this triangulated surface, we apply our method. First, the dual graph of this surface is determined and we apply the placement algorithm on that graph to determine a suitable placement of the pantograph mechanisms. When the mechanisms are placed on the surface according to this placement, we obtain the shape shown in Fig. 9b and Supplementary Movie 2.

For the cardioid we have constructed here, the rotation angle $\theta$ can lie in the range $[-20.5°, 20.3°]$, resulting in a maximum scaling factor of $\lambda = 0.937$. A movie of the resulting dilational surface moving between its extremal points is included in the supplementary materials.

As a final example, we have taken the Stanford Bunny[32], shown in Fig. 10a. For the bunny, we took an available triangulation[33], and edited the triangulation manually to remove the triangles with the sharpest angles in order to increase the maximum scaling factor. The resulting model is shown in Fig. 10b. This mesh was then fed into our algorithm, which computed a suitable placement of the pantograph mechanisms. The resulting dilational mechanism is shown in Fig. 10c and Supplementary Movie 3.

For this mechanism, the rotation angle $\theta$ can lie in the range $[-15.0°, 13.1°]$, resulting in a maximum scaling factor $\lambda = 0.966$. This scaling factor is not limited by the shape of the Stanford Bunny, but rather by the specific triangulation used to approximate it and the placement of the pantographs on the triangulation. The maximum scaling factor could be increased further by triangulating the Stanford bunny such that the triangles are close to equilateral, thereby removing even more sharp angles from the polyhedron. Even so, the linear scaling of 3.4% obtained here is already similar to the diametric expansion of human arteries during the cardiac cycle[34].

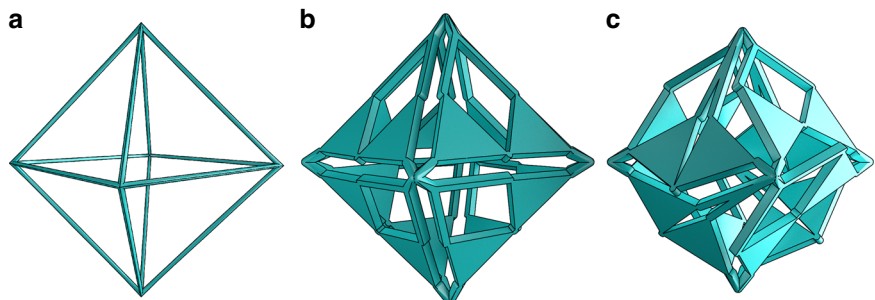

**Fig. 7** Dilation of an octahedron. Eight pantograph linkages are placed on the octahedron. **a** shows the wireframe of an octahedron. **b** shows the dilational structure in the neutral position, ($\theta = 0$) and (**c**) shows a compacted position ($\theta = 25°$)

## Discussion

In this work, we introduce a comprehensive strategy to achieve dilation of any surface. We do this by triangulating the surface and replacing the triangular faces with Sylvester's pantographs. The similarity points of this pantograph always span a similar triangle. We constrain these triangles in such a way that their orientation is preserved, this preserves the shape of the triangulated surface while allowing it to scale.

Kinematically, a structure constructed using this strategy can be scaled to a single point from its original size. In practice, however, the range of motion of the pantographs is limited by both collisions between links within a pantograph, and collisions between neighboring pantograph mechanisms.

Collisions within the pantograph linkage cause the pantograph with the smallest range of motion to limit the motion of the whole structure, since all pantographs share a single degree of freedom.

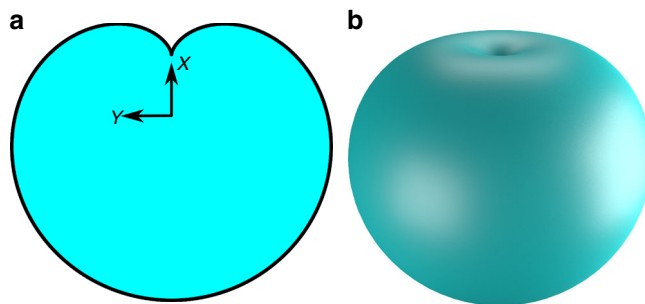

**Fig. 8** The cardioid surface. The cardioid surface is constructed by revolving the planar cardioid curve. **a** shows the curve, **b** shows the complete revolved cardioid

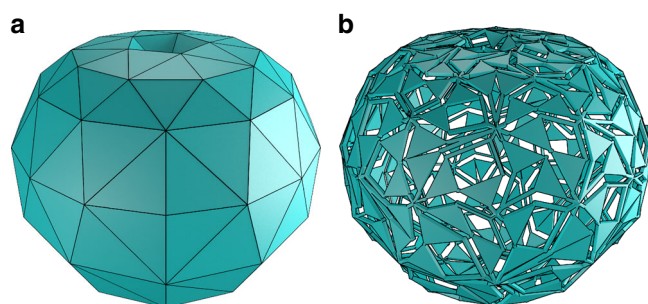

**Fig. 9** Dilation of a cardioid. The surface of the cardioid is first triangulated (**a**), after which each triangular face is replaced by a skew pantograph (**b**) to obtain a dilational surface. A movie of the final mechanism is included in the supplementary material

This could be improved by changing the triangulation strategy and optimizing the placement of the pantographs such that the top-angle of the triangles comes out more favorable (possibly favoring one motion direction over the other). Better pantograph placements might be found, since our pantograph placement algorithm yields non-unique solutions to the pantograph placement problem.

We avoid collisons between neighboring pantograph mechanisms by tiling them in a specific manner. We used a graph-based approach to generate suitable placemeents of mechanisms and showed that this approach works for any triangulated surface.

We have illustrated our method with three examples: an octahedron, a cardioid and the Stanford bunny. These surfaces increase in complexity and have varying Gaussian curvature. For the octahedron, the maximal scaling and suitable tiling can be determined by hand. For the cardioid and the Stanford bunny, there are many more triangular faces and the faces are more irregular, for which we present computational methods to generate dilational structures for these surfaces.

The planar kinematics of the pantographs ensure that the resulting dilational mechanism stays close to the described surface throughout the range of motion. This leaves the encompassed interior entirely empty.

A interesting side-effect is that our implementation of the method is directly compatible with the often used STL file format for 3D objects. As such, our strategy could be implemented as a one-click solution to create dilational models.

## Methods

**STL preparation**. The STL files for the octahedron and the cardioid were created using OpenSCAD and Matlab, respectively. For the Stanford Bunny Model, the original STL files were prepared using OpenSCAD and Matlab. For the Stanford Bunny model, the original triangulated model by Thingiverse user johnny6[33] was adjusted in Blender to decrease the number of sharp corners in the triangular faces.

The used STL files have been made available in the Supplementary Data.

**Algorithm implementation**. The algorithm described in Supplementary Note 2 was implemented in Python3, making use of the NetworkX[35] and numpy-stl modules.

First, the prepared STL files were read using the numpy-stl module and the faces, points and edges were extracted. From this information, the dual graph of the triangulated structure was constructed. On this graph, the tile placement algorithm was run to obtain a suitable placement of the pantographs.

The STL files were processed using Python3, using the STL module, the faces, points and edges were extracted. This information was then used to create dual graph of the triangulated structure. On this graph, the triangle placement algorithm, as described in Supplementary Note 2, was run to obtain a suitable placement of the pantographs. Using this placement, two lists were created, one containing the three vertex positions for each face, always starting with the top vertex of the pantograph, the other indicating the motion direction for the face. These lists were finally used to create 3D models of the tiled structures in OpenSCAD.

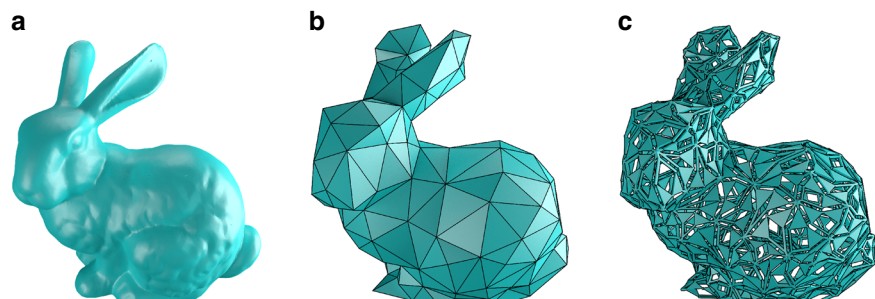

**Fig. 10** Dilation of the Stanford bunny. **a** shows the original Stanford bunny[32]. **b** shows our adaptation of the triangulated version by Thingiverse user johnny6[33], which was used to create the dilational surface shown in **c** by replacing each triangular face with skew pantograph mechanisms. The resulting surface has a scaling factor of $\lambda = 0.966$. A movie of the final mechanism is included in the supplementary material

**Movie generation**. Using the models of the dilational structures, movies were created by rendering the models for a range of rotation angles in OpenSCAD. The rendered frames were then combined into a movie using FFMPEG.

## Data availability

The datasets generated during and/or analyzed during this study are available in the 4TU. ResearchData repository, DOI: 10.4121/uuid:36cfec67-aa04-469f-ac13-64e7d95a0c18.

## Code availability

Algorithms used in this study are included in this published article (and its supplementary information files). Code will be made available from the corresponding author upon reasonable request.

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

## Acknowledgements

We thank M.P. Noordman (University of Groningen) for assistance with the formulation of the graph theory proof. The work was part of the Nanoscience Engineering Research Initiative of TU Delft. The authors would like to acknowledge NWO-TTW (HTSM-2012 12814: ShellMech) for the financial support of this project.

## Author contributions

F.G.J.B. and W.W.P.J.S. proposed and designed the research, performed the numerical calculations and wrote the paper. F.G.J.B. wrote the algorithm. J.L.H. proposed the cardioid example. V.W. and J.L.H. supervised the project and reviewed the paper.

## Competing interests

The authors declare no competing interests.
