## [Peer Review File · Nature Communications]

Reviewers' comments:

Reviewer #1 (Remarks to the Author):

The paper is OK but seems not to be particularly important. I have the impression it was written with a view to publishing something in Nature, rather than to address any important problem. I am neutral on its suitability for publication.

Reviewer #2 (Remarks to the Author):

This paper introduced a new strategy to achieve dilation of shape or surface and triangulating the surface and replacing the triangular faces with pantographs. Theory and numerical modeling are well organized, and readers of nature communications are likely to be interested. However, I would like to share the following comment.

- The author said that this method could be applied to "any shape", which seems a bit dangerous. In other words, it seems to be a little difficult to generalize with the models shown as examples in the manuscript.

In models with singular point (such as sharp surfaces), it seems necessary to discuss how this method works.

Reviewer #3 (Remarks to the Author):

The authors present a general method for creating dilational structures with arbitrary shape, where all motions are tangent to the described surface.

This method consists of triangulating the target curved surface and replacing each of the triangular faces by pantograph mechanisms according to a tiling algorithm that avoids collisions between neighboring pantographs.

The paper is interesting and deserves publication. However, a better attention on the mechanical characterization of pantographs (since it is well noted in the recent literature) could be considered by the authors. For example, at line 87, after "pantograph" the authors could insert a reference on the mechanical characterization of such pantographs, adding ", see also pantographic beam or structures [A] and references therein," with "[A] BOUTIN, Claude, et al. Linear pantographic sheets: Asymptotic micro-macro models identification. Mathematics and Mechanics of Complex Systems, 2017, 5.2: 127-162."

Reviewer #1 (Remarks to the Author):

The paper is OK but seems not to be particularly important. I have the impression it was written with a view to publishing something in Nature, rather than to address any important problem. I am neutral on its suitability for publication.

We highlighted the novelty and importance of our method. First, it is useable on any surface (i.e. 2-manifold); this is not seen in the literature up to this point. Second, the mechanisms are only located on this surface and not (significantly) in the volume above or below it as with other solutions.

Dilational structures with the above properties could be of use in, for instance, structures that grow with a person such as medical braces for children, expandable furniture, medical devices that require stowability or compression but need to be stiff otherwise, or implants that need to accommodate some motion but maintain their shape, such as aortic stents. These are some examples of applications; however, the manuscript was not written with a specific application in mind. Rather it was written to prove that a general method exists for every type of surface and that the topology of such a structure need not be changed according to the shape of that surface. We are pleased though to include relevant applications in the paper to show the importance of our work, these additions can be found in the introduction of the manuscript.

Reviewer #2 (Remarks to the Author):

This paper introduced a new strategy to achieve dilation of shape or surface and triangulating the surface and replacing the triangular faces with pantographs. Theory and numerical modeling are well organized, and readers of nature communications are likely to be interested. However, I would like to share the following comment.

- The author said that this method could be applied to “any shape”, which seems a bit dangerous. In other words, it seems to be a little difficult to generalize with the models shown as examples in the manuscript.

In models with singular point (such as sharp surfaces), it seems necessary to discuss how this method works.

We would like to thank the reviewer for pointing this out; any shape is incorrect. Any mention of shape is changed to any surface (any 2-manifold with or without a boundary). In this case the triangulation ensures

that any edge is shared by at most two triangles, which corresponds with our graph-based method. We stress this fact in section 2.2 and 3.2.

Geometries where surfaces intersect violate this condition and might have edges that are shared by more than two triangles. Our method is not explicitly suited to solve such problems.

Sharp surfaces with singular points are not an issue as these are still 2-manifolds that after triangulation still have edges that are shared by at most two triangles. Take for instance the octahedron, this polyhedron can be stretched along a single vector. If this stretching is taken to the extreme, it will result in a very sharp surface. However, this would not change the topology of the polyhedron and therefore not affect the tiling solution. The same reasoning is valid for any triangulated surface.

Reviewer #3 (Remarks to the Author):

The authors present a general method for creating dilational structures with arbitrary shape, where all motions are tangent to the described surface.

This method consists of triangulating the target curved surface and replacing each of the triangular faces by pantograph mechanisms according to a tiling algorithm that avoids collisions between neighboring pantographs.

The paper is interesting and deserves publication. However, a better attention on the mechanical characterization of pantographs (since it is well noted in the recent literature) could be considered by the authors. For example, at line 87, after "pantograph" the authors could insert a reference on the mechanical characterization of such pantographs, adding ", see also pantographic beam or structures [A] and references therein," with "[A] BOUTIN, Claude, et al. Linear pantographic sheets: Asymptotic micro-macro models identification. Mathematics and Mechanics of Complex Systems, 2017, 5.2: 127-162.

We have elaborated the description of the pantograph in section 2.3. We discuss the geometric properties that enable the dilational behaviour of the skew pantograph. Please see the highlighted section in subsection 2.3 of the adjusted manuscript. As a reference we used a book that describes the characterization of the skew pantograph in detail [Hall1961].

REVIEWERS' COMMENTS:

Reviewer #2 (Remarks to the Author):

This paper demonstrated a method to create mechanism-based dilational structures fitting to a spatially curved surface of 2-manifolds.

The reviewer's comments are reflected well in the revised manuscript, and it would be interesting to the readership of Nature Communications.